# Lateral Deformation of Human Red Blood Cells by Optical Tweezers

**DOI:** 10.3390/mi12091024

**Published:** 2021-08-27

**Authors:** Pavel Yale, Michel A. Kouacou, Jean-Michel E. Konin, Eugène Megnassan, Jérémie T. Zoueu

**Affiliations:** 1Laboratoire de Physique Fondamentale et Appliquée, UNA, Abidjan BP 801, Côte d’Ivoire; kedoukoua@yahoo.fr (J.-M.E.K.); megnase@yahoo.com (E.M.); 2Laboratoire d’Instrumentation, Image et Spectroscopie, INPHB, Yamoussoukro BP 1093, Côte d’Ivoire; abakaci@yahoo.fr (M.A.K.); jzoueu@yahoo.fr (J.T.Z.)

**Keywords:** optical tweezers, human red blood cells, indentation, shear modulus

## Abstract

In this paper, we studied the lateral deformation of human red blood cells (RBCs) during lateral indentation by an optically trapped silica bead with a diameter of 4.5 µm (Bangs Laboratories, Inc. Fishers, IN, USA). The images were captured using a CCD camera and the Boltzmann statistics method was used for force calibration. Using the Hertz model, we calculated and compared the elastic stiffness resulting from the lateral force, showing that the differences are important and that the force should be considered. Besides the lateral component, this setup also allowed us to examine the lateral cell–bead interaction. The mean values of the cell shear stiffness measured during indentation were 3.37 ± 0.40 µN/m for biconcave RBCs, 3.48 ± 0.23 µN/m for spherical RBCs, and 3.80 ± 0.22 µN/m for crenelated RBCs, respectively. These results show that this approach can be used as a routine method for RBC study, because it enabled us to manipulate the cell without contact with the wall.

## 1. Introduction

The manipulation of biological specimens using light microscopy is a subject of increasing interest due to its applicability and relevance to fundamental research [1]. The demonstration that Optical Tweezers (OT) are able to conveniently manipulate a single living cell as a noninvasive trapping method was an important achievement for biology [1,2]. The advantages of OT compared to atomic force microscopy (AFM) and micropipette aspiration are their higher sensitivity, their simple control of the laser light intensity to adjust the applied force with an accuracy down to piconewtons, and the lack of risk of contamination. OT have made a drastic contribution to the natural sciences, offering abundant opportunities to study micro- and nano-sized objects in biology and physics. OT enable the noninvasive manipulation of biomolecules (protein and DNA), viruses, bacteria, and cells and have been proven to be safe for living objects. The application of OT is also promising in the blood rheology field. In recent years, the vital regulatory role of biological forces in embryogenesis has been gradually recognized and valued, especially in cardiovascular system development [3]. The biomechanical properties of red blood cells (RBCs) arise from their unique deformability, which enables them to travel through the human body’s smallest capillaries. Being relatively simple structures of the phospholipid bilayer membrane and lacking nuclei and some organelles, RBCs are often used as a model system for cell mechanodynamics studies, revealing the interrelation between structural, biological, and chemical signals and the corresponding mechanical response [4]. Furthermore, the microvascular system requires RBCs to undergo significant cellular deformation in order to pass through vessels whose diameters are significantly smaller than their own [5]. The ‘biconcave’ shape of the RBC is transformed into a ‘bullet’ shape during the flow of blood through small capillaries, and the cell fully recovers its original shape when the constraint or loading causing the shape change is released [6]. Two glass or polystyrene beads attached to a human RBC are manipulated by two focused laser beams to stretch the cell and study its mechanical characteristics [5,7]. Infrared optical tweezers are used to trap and manipulate RBCs within subdermal capillaries in living mice [8]. An OT study of red blood cell aggregation and disaggregation in plasma and protein solutions has been performed [9]. The local elasticity of HBL-100 cells, an immortalized human cell line originally derived from the milk of a woman with no evidence of breast cancer lesions, was studied using optical tweezers [10]. Recently, Konin et al. carried out a dynamics study of the deformation of RBCs using OT [11]. Cell elasticity can be locally measured by pulling membrane tethers or stretching or indenting the cell using OT. In the indentation method, most of the cells studied attach to the slide surface. However, it is easier to find both the mobile cells and the silica beads in the liquid medium. Therefore, to reduce the experimentation time, we propose in this paper an approach to performing cell indentation by laterally moving the RBC against the trapped silica bead. Since the trap position is fixed, the trapped silica bead displacement directly reflects its interaction with the RBC. The development of and future prospects for the application of OTs in hemorheology, including practical in-depth studies of RBC formation for therapeutic, functional, and diagnostic needs, are highlighted.

## 2. Materials and Methods

### 2.1. Experimental Setup

In this section, we describe in detail the equipment and controls used for the experiment. Figure 1 shows the experimental setup. The setup is based on the modular Thorlabs OT kit (OTKB, Thorlabs Inc., Newton, NJ, USA). The OT setup consisted of a Diode Laser (PL980P330J) at a wavelength of 980 nm with an output power of up to 330 mW. A large numerical aperture (NA 1.25) Nikon 100x oil immersion objective (MRP01902, Nikon, Tokyo, Japan) was used to focus a laser beam and form an optical trap. A white light-emitting diode (LED) source was mounted above the optical trap in order to illuminate a sample with light in the visible part of the electromagnetic spectrum. The forward-scattered light transmitted from the sample was collected by a Nikon 10x air condenser. The sample was mounted on a 3-axis piezo translation stage (MAX301) with strain gauge feedback. The particles used in these experiments were silica beads with diameters of 4.5 µm (Bangs Laboratories, Inc., Fishers, IN, USA) and human RBCs. In order to prepare a sample for our experiment, the RBCs were separated by centrifugation and suspended in different physiologically relevant mediums, such as 150 mOsm, 300 mOsm, and 900 mOsm phosphate buffered saline (PBS). Each solution was incubated with silica beads with 4.5 µm diameters. The images of the RBCs and silica beads were captured using a CCD camera and recorded on a videotape. The video images were then downloaded onto a computer and digitized for image analysis. The individual frames of the recorded movies were analyzed using the Image-J software (version 1.43, National Institutes of Health, Bethesda, MD, USA). The laser power measured after the objective was used in our experiment covered a range from 62 mW to 76 mW.

### 2.2. Method and Force Calibration

The cell elastic stiffness was measured locally by horizontal indentation using a trapped silica bead as a probe. Before the measurement, the silica bead was trapped and a video in which there was no contact between the trapped silica bead and the cell was recorded. From this video, the force which maintained the silica bead in the trap was calculated.

The experimental approach is shown in Figure 2. The silica bead was optically trapped above the cell, without touching it. Then, the stage was displaced horizontally so that the cell came into contact with the trapped silica bead. When the cell intercepted the bead, it exerted a force causing the displacement of the bead from the trap equilibrium position. The bead also began to push the cell, inducing an indentation *Hc* in the cell membrane.

Each time the distance between the silica bead and the RBC decreased, the indentation force increased and the deformation became significant.

It is possible to measure the bead movement into the cells (the indentation *Hc*) by the following relation:(1)$$Hc=\frac{1}{2}\left(D-\sqrt{D^{2}-D_{i}^{2}}\right),$$
where D is the silica bead diameter and $D_{i}$ is the measured mean diameter of the indentation in micrometer (see Figure 2).

Another parameter required to calculate hardness and elasticity is the contact force between the cell and the trapped bead. The trapped bead, initially trapped, is found in a position of stable equilibrium. Since a stable equilibrium corresponds to a potential well, the method of Boltzmann’s statistics can be used to calculate the induced forces. At equilibrium, the potential well of the trapped bead is reconstructed with a potential written as: $U_{1}=k_{1}{\left(\Delta r_{1}\right)}^{2}/2$. The interaction of the RBC with the trapped bead will then disturb the first equilibrium position of the trapped bead. The trapped bead will have a new equilibrium position in this new phase, with a potential well $U_{1}^{\prime }=\frac{k_{1}^{\prime }{\left(\Delta r_{1}^{\prime }\right)}^{2}}{2\;}\;\mathrm{with}\;k_{1},\;\;k_{1\;}^{\prime }et\;r_{1\;},r_{1}^{\prime }$ representing the trap stiffness and the position of the trapped bead before and after the interaction, respectively. Since it is the trapped bead that we can master, the energy communicated to the trapped bead by the RBC can then be written [12]:(2)$$U=U_{1}-U_{1}^{\prime }=\frac{k_{1}{\left(\Delta r_{1}\right)}^{2}}{2}-\frac{k_{1}^{\prime }{\left(\Delta r_{1}^{\prime }\right)}^{2}}{2}.$$

As the force derives from the potential F=gradU, the interaction force can be written according to [12].

Video analysis was used to locate the silica bead center of mass for each frame and reconstruct the silica bead path. Image-J software was used to reconstruct the *xyz* path of the silica bead from a video recording of its Brownian movement. Several independent methods that have been used to determine the trapping force based on the Brownian motion of a trapped particle [13,14]. However, in this study we used Boltzmann’s statistics to obtain the trap stiffness. Using Boltzmann’s statistics, the optical potential reconstruction can be used to calculate any continuous trapping landscape in the accessible region by thermal agitation [13]. In equilibrium, the probability density *p*(*x*) of the 1D particle position is given by:(3)$$p\left(x\right)dx=Ce^{\frac{-E\left(x\right)}{K_{B}T}},$$
where *C* is the normalization constant and *E*(*x*) is the trap potential. The shape of *E*(*x*) can be determined from the normalized histogram of the trapped bead positions as:(4)$$\begin{array}{c} E\left(x\right)\;=\frac{1}{2}k_{trap}\left(x{)}^{2}=-k_{B}T\;ln(p\left(x\right)\right),\\ k_{trap}=\frac{-2k_{B}T}{x{\left(t\right)}^{2}}\;ln(p\left(x\right)). \end{array}$$

In the case of the commonly used TEM_00_ Gaussian trapping beam, which results in a harmonic trapping potential, one can fit a parabola $y=ax^{2}+b$ to the data in the central region of the potential to extract the trap stiffness and check for possible deviations from the perfect harmonic shape. The trap stiffness *k_trap_* = 2*a*/*k_B_T* obtained in such a manner is more accurate than Equation (4). Another advantage of such calibration is that it also provides information about the potential in the region away from the trap center, where the optical potential is non-harmonic [13].

### 2.3. Elastic Stiffness and Shear Stiffness Calculation

From the measurements of indentation and force, the Hertz model was used to obtain the elastic modulus [15]. Although this model applies to homogeneous, semi-infinite elastic solid objects, a living cell is clearly different from that type of object, being viscous as well as elastic and inhomogeneous [10]. In fact, since the goal of most experiments is to perform comparative studies between cells under different environmental conditions or between different cells, the use of the Hertz model can be justified. In our experiments, we used the Hertz model to obtain the elastic modulus [15]. The elastic stiffness, *Eh* is given by:(5)$$Eh=\left[\frac{3\left(1-\nu^{2}\right)}{4\;\cdot \sqrt{H_{c}R}}\right]\cdot F,$$
where R is the microbead radius, F is the force, $H_{c}$ is the indentation, and *ν* is the Poisson ratio. For these experiments, we used ν=0.5. In the literature, the cortical shear stiffness *Gh* is usually given rather than the elastic stiffness *Eh*. The quantities are related by:(6)$$Gh=\frac{Eh}{2\left(1+\nu \right)}.$$

## 3. Results

### 3.1. Mechanical Properties of Biconcave RBCs

We used a bead with a diameter of 4.5 µm as the indenter. This bead was trapped with a power of 71.05 mW. For each variation in the contact diameter, the force applied is given. Figure 3 presents the various images resulting from the videos recorded at each measurement. In this study, the bead displacement was horizontal (*x* direction); the force had only one component, *Fx* (Figure 3); and the indentation, *Hx*, was only horizontal.

Using the Hertz model, we then calculated the elastic stiffness corresponding to the contact force *F.* The value of maximum force, when the diameter of indentation did not vary any more, was 30.52 pN, with a height of depression of 1.279 µm. The properties obtained from the RBC, elastic stiffness and the shear stiffness, were *Eh* = 11.08 µN/m and *Gh* = 3.69 µN/m, respectively. The mean values of these properties were *Eh* = 10.11 ± 1.20 µN/m and *Gh* = 3.37 ± 0.40 µN/m.

### 3.2. Measurements Obtained from Spherical and Crenelated RBCs

We also used a silica bead with a diameter of 4.5 µm. This bead was trapped with a power of 69.05 mW. The force applied is given for each variation in the contact diameter. The various images resulting from the videos recorded with each measurement are presented in Figure 4.

The mean values of the properties obtained, in particular the elastic stiffness and the shear stiffness, were *Eh* = 10.46 ± 0.70 µN/m and *Gh* = 3.48 ± 0.23 µN/m for the spherical RBCs and *Eh* = 11.40 ± 0.67 µN/m and *Gh* = 3.80 ± 0.22 µN/m for the crenelated RBCs. Table 1 presents all the measurements of the elasticity modulus and the shear modulus obtained from these three types of RBCs.

The results presented in the Table 1 show that RBCs with a biconcave form were more elastic than the spherical and crenelated RBCs.

## 4. Discussion

A normal erythrocyte presents a profile like a biconcave disc containing hemoglobin. This form confers a significant elasticity upon it, which allows the transport of dioxygen through certain narrow capillaries. The results presented in Table 1 show that the biconcave RBCs are more elastic than spherical and crenelated RBCs. This difference can be explained by the difference in morphology between the RBCs due to the physiological medium. When the RBC deviates from the biconcave form, it becomes fragile, less flexible, and can no longer circulate in small capillaries. This is also explained by natural dispersion. During their lifespan, proteins are adsorbed on the surface of the RBC membranes, which causes a loss of elasticity. Moreover, the number of skeleton defects increases with the age of the RBCs [16]. When the RBCs become too rigid, they can no longer pass through the spleen, which becomes blocked and is eventually destroyed.

The shear stiffness values calculated for biconcave, spherical, and crenelated RBCs are of the same order of magnitude as those obtained by the method of aspiration in a micropipette [17] (4 < *Gh* < 10 μN/m), only inferior. These same values are also of the same order of magnitude as those obtained by *Gh* = 1.233 µN/m [18] and *Gh* = (2.4 ± 0.4) µN/m [19], only higher. We explain this difference by the population selection of different RBCs according to their affinity with glass. Aspiration experiments with non-glass micropipettes do not function for membranes that adhere to micropipettes. Measurement is thus performed on RBCs that adhere to glass.

In certain cases of OT, a category of RBC that has a great affinity with glass is used. With this technique, two beads of glass adhere spontaneously to the RBC membrane. Affinity with glass can depend on the surface quality of the membrane, which is connected to the density of proteins adsorbed on the membrane. However, the density of proteins grows with the RBC age [20]. Similarly, the elastic properties depend on the age of the RBC.

## 5. Conclusions

In this study, we used a simple OT setup to measure RBC–bead interaction by the lateral displacement of the RBC against a trapped bead. The Hertz model was used to calculate the lateral indentations resulting from lateral forces.

The use of this indentation technique enabled us to manipulate the cell without it losing contact with the laser. OT is easy to set up and can thus help us to obtain the physical properties of RBCs arising from indentation. This technique can be used in the future to characterize infected RBCs.

## Figures and Tables

**Figure 1 micromachines-12-01024-f001:**
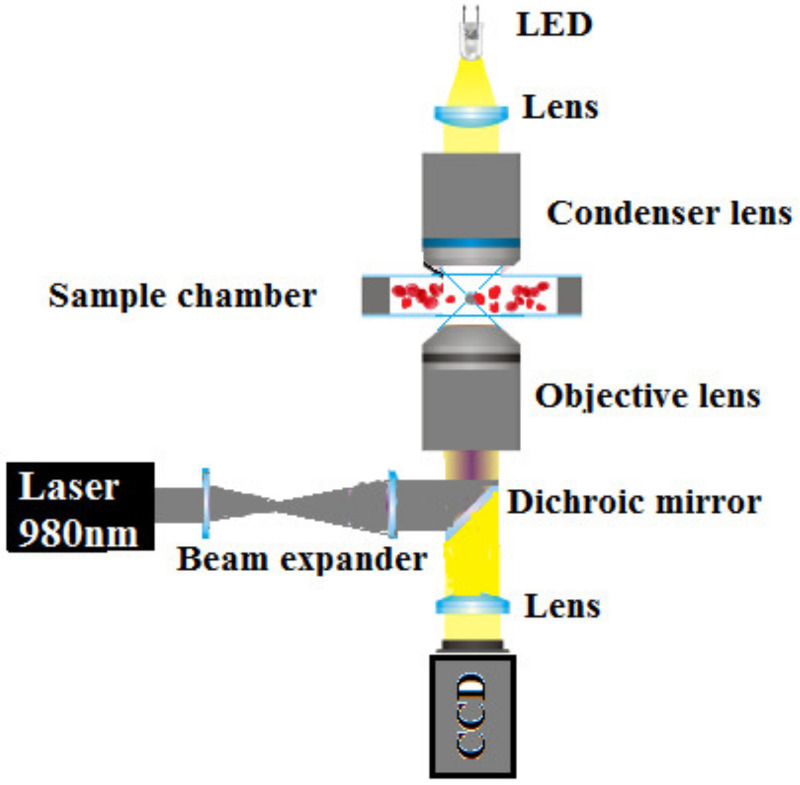
Scheme of the experimental setup.

**Figure 2 micromachines-12-01024-f002:**
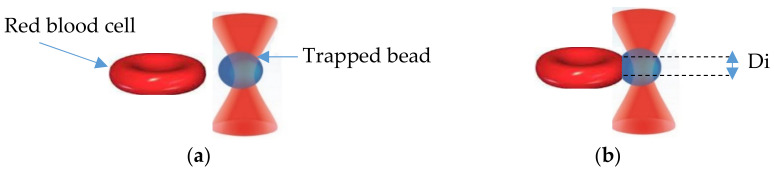
Interaction illustration: (**a**) no interaction; (**b**) interaction between RBC and trapped silica bead.

**Figure 3 micromachines-12-01024-f003:**
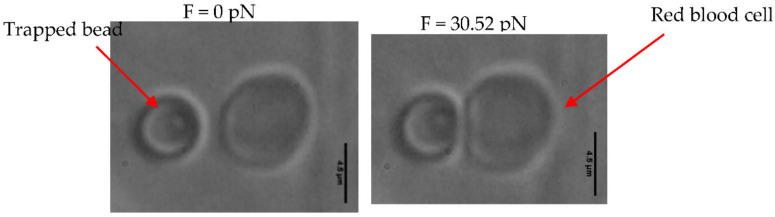
Images of the silica bead interacting with the biconcave RBC in PBS solution with an osmolarity of 300 mOsm. Ten biconcave RBCs were used. The scale bar represents 4.5 μm.

**Figure 4 micromachines-12-01024-f004:**
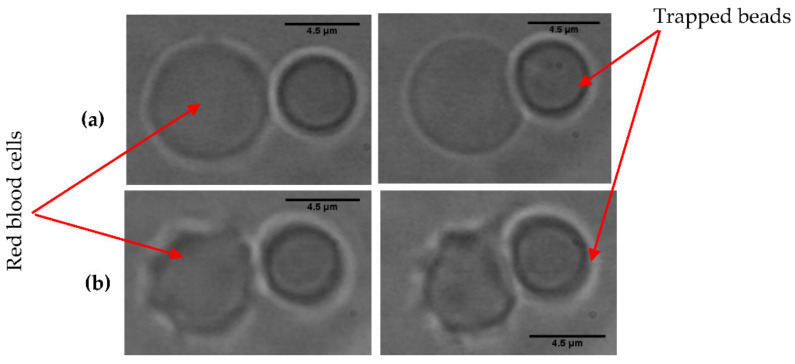
(**a**) Image of the interaction between the trapped bead and the spherical RBC in PBS solution with an osmolarity of 150 mOsm, (**b**) image of the interaction between the crenelated RBC and the bead in PBS solution with an osmolarity of 900 mOsm. Ten spherical RBCs and 10 crenelated RBCs were manipulated.

**Table 1 micromachines-12-01024-t001:** Mechanical properties values of the studied RBCs.

Parameters	Biconcave RBCs	Spherical RBCs	Crenelated RBCs
Elastic stiffness (µN/m)	10.11 ± 1.20	10.46 ± 0.70	11.40 ± 0.67
Shear stiffness (µN/m)	3.37 ± 0.40	3.48 ± 0.23	3.80 ± 0.22

## Data Availability

Data can be made available upon request to the corresponding authors.

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
