# Peer review of "Lateral Deformation of Human Red Blood Cells by Optical Tweezers"

_micromachines, 2021, doi:10.3390/mi12091024_

Round 1

Reviewer 1 Report

Dear authors,

The reviewers appreciate your effort to elaborate a complex methodology to determine the  elastic characteristics of various structures of red blood cells (RBC).

Because in your investigations have been developed some mechanical equations, please accept our observations:

  1. Eq. (1) is not presented in reference [12] and Di is not presented in Fig. 2. If Di is the measured mean diameter of the indentation circular surface, a simple Pythagoras relation leads to another relation for Hc. From our calculus Hc can be approximated by relation Hc=Di2/8D.
  2. The authors propose the Hertz model to determine the Elastic modulus and shear modulus for RBC. The reviewer warn the authors that in Hertz contact the relations between elastic deformations and applied forces are not linear relations. In reference [12] sugested by authors for Eq. 5  the Elastic modulus E is obtained by nonlinear relations. Can the authors explain how they got the relation (5)?
  3. In reference [12] elastic modulus E is expressed in Pa or N/mas is normal in mechanical applications. In the present manuscript the  elastic modulus and shear modulus are expressed as the stiffness parameters expressed in µN/m.
  4. In Conclusions  the authors explain that used  the "linearized Hertz model". The reviewer suggest to the authors to explain  how they "linearized" the Hertz model.
  5. In the context with above comments the values from the Table 1 are not Elastic and shear modulus, they are the normal and tangential stiffness, if the relations used are correct!!!
  6. I suggest to the authors more details in development of the mechanical relationships.

Author Response

Response to reviewer 1 comments

1) Eq.(1) is notpresented in reference 12 and Di is not presented in fig.2. if Di is the measured mean diameter of indentation circularsurface, a simplepythagoras relation leads to another relation for Hc can be approoximated by relation Hc=Di2/8D.

Response 1: Thank you for bringing this inconsistency to our attention. We deleted the reference 12 for this Eq(1) and presented Di in Fig.2.

Reviewer 2 Report

At the moment, the paper is weak and does not convey any significance to this topic. I have the following suggestions to improve the quality of the paper to be further considered for publication. 

  • What is OT? First, mention the full form and then use the abbreviation.
  • The main aim of the work is not clear. I suggest the author extend the introduction section to explain in detail the motivation behind this research. And where it can be employed in practical life?
  • It should be mentioned that at which laser power, the RBC can be damaged?
  • In figure 3 and figure 4, label the RBC and silica bead.
  • I suggest the author improve the English language used in the paper. Some details are not clear. Moreover, the language used in the paper is not purely scientific. Even the first line of the abstract “This paper, we studied the lateral deformation of human red blood cell (RBC) during the lateral indentation by an optically trapped silica bead of diameter 4.5 μm (bangs laboratories, Inc).” is not in a proper construction.
  • The conclusion section should be modified and mention the main results in this section.
  • Is it possible to maneuver the RBC with this optical tweezer? Please explain.
  • I suggest the author give a list of previous studies on optical tweezers and highlight the points in which the proposed research is better than the previous results.

Author Response

Response to reviewer 2 comments

Round 2

Reviewer 1 Report

Dear authors,

The reviewer observe your modification of the initial observation and, consider that, in general can be accepted. So, Eq. 1 can be considered acceptable because the cell indentation Hc<<D.

Reefer to the Eq. (5), in this form it leads to an elastic stiffness Eh  because is expressed in µN/m. Also it can be considered that you modified  Eq. (1) from Reference [12] considering Hc1/2 instead of Hc3/2 that results from Hertz model. With this modification you obtained for Eh the units similar to an elastic stiffness and not to a classical elastic modulus ( named Young modulus).   It can be consider that your  relation of Eh is a first approximation for elastic properties of the red cells.

The reviewer accept the manuscript in the modified version.  

Reviewer 2 Report

I am willing to accept the paper in its current form.

This manuscript is a resubmission of an earlier submission. The following is a list of the peer review reports and author responses from that submission.